# Psychological and financial impacts of COVID-19-related travel measures: An international cross-sectional study

**Pippa McDermid[1], Soumya Sooppiyaragath[2], Adam Craig[1], Meru Sheel[3], Katrina Blazek[1], Siobhan Talty[4], Holly Seale[1]***

**1** School of Population Health, Faculty of Medicine and Health, University of New South Wales, Sydney, New South Wales, Australia, **2** Independent Research Psychologist, Canberra, Australia, **3** Sydney School of Public Health, Faculty of Health and Medicine, the University of Sydney, Sydney, New South Wales, Australia, **4** Freelance Monitoring, Evaluation, Accountability and Learning Technical Advisor, The ME(AL). T.A., Granada, Spain

* h.seale@unsw.edu.au

**Data Availability Statement:** All data files are available from the Figshare database; the DOI is 10.

## Abstract

### Background

The impact of COVID-19 international travel restrictions has to date, not been fully explored, and with the ongoing threat that new variants could potentially restrict movement further, it is important to consider the impacts that travel restrictions have on community members. This study aimed to evaluate the psychological and financial impact of COVID-19 travel restrictions on those separated from their partners or immediate families, as well as temporary visa holders who were unable to migrate.

### Methods

Between 4 November 2021 to 1 December 2021, we executed a cross-sectional online survey targeting three specific groups; (1) those stranded from their partners; (2) those stranded from immediate families; and (3) temporary visa holders unable to migrate or cross international borders. We collected data on respondents' demographic profile; the nature of COVID-19-related travel impacts; depression, anxiety, and stress levels (using the validated DASS-21); and finally, data on respondents financial, employment and accommodation situation.

### Results

1363 respondents located globally completed the survey. 71.2% reported financial stress, 76.8% ($\bar{x} = 11.57$, SD = 5.94) reported moderate-to-extremely severe depression, 51.6% ($\bar{x} = 6.80$, SD = 5.49) moderate-to-extremely severe anxiety, and 62.6% ($\bar{x} = 11.52$, SD = 5.55) moderate-to-extremely severe stress levels. Statistically significant factors associated with moderate-to-extremely severe depression, anxiety, and stress included being female, chronic illness, and experiencing financial stress. Employment during COVID-19 —specifically essential services workers or unemployed—was associated with higher levels of

6084/m9.figshare.19470260 (https://figshare.com/articles/dataset/Travel_Restriction_Impacts_Survey_Data_sav/19470260).

**Funding:** The author(s) received no specific funding for this work.

**Competing interests:** The authors have declared that no competing interests exist.

anxiety and stress, with only essential workers being a predictor of higher stress severity. Factors that provided psychological protection included being older and having children.

## Conclusion

This study is one of the first to explore the impact COVID-19-related international travel restrictions have had on the financial status and psychological health of affected individuals. It highlights the significant human cost associated with the restrictions and identifies psychologically vulnerable populations. These results will help the design of targeted health and social policy responses.

## Introduction

International mobility has changed dramatically in the wake of the COVID-19 pandemic.

The rapid introduction of international travel restrictions in early 2020 resulted in a huge downward shift in cross-border movements of individuals globally with many people finding themselves stranded abroad [1–4]. At the end of the second year of the pandemic (December 2021), some countries continued to maintain strict international border controls, with frequent changes to travel and border-control policies. In response to the omicron variant, a proportion of countries had reinstated a ban on travellers from high-risk countries, others continued to impose total border closure, or stringent quarantine/screening requirements on international arrivals [5]. Furthermore, with new COVID-19 variants expected, the level of uncertainty about international travel restrictions continues and border controls will likely remain a frequently used emergency response [6].

Major economic concerns have been raised globally about the impact of travel restrictions, specifically those denying entry to tourists, skilled workers and international students, due to the impacts this will have on country income, productivity, labour supply, university funding and growth [7–9]. Beyond the negative economic effects of denying entry to these groups, are the potential psychological and financial impacts on the individual [10]. Research thus far has explored the impact of COVID-19 on short-term migrant workers within the country that they are working, and found that this group is one of the most vulnerable to unemployment, depression, food insecurity and homelessness [11, 12].

In addition to the skilled workers and visa holders who have been unable to enter certain countries, travel restrictions have resulted in the forced separated of families and partners during the pandemic, increasing the risk of social isolation. Stories emerged in both the social and mass media channels, of children being separated from their parents [13], as well as family members being unable to travel to visit sick and elderly family members [14].

Current evidence has focused on the impacts of lockdowns and social distancing measures on romantic relationships, with findings reflecting increasing rates of loneliness, social isolation and negative impacts on the general health and well-being of populations globally [15].

Previously our team examined the psychological and financial distress reported by citizens and permanent residents stranded abroad due to international travel restrictions introduced in response to the COVID-19 pandemic. This study was open to any individual stranded abroad from their country of citizenship or permanent residence due to international travel restrictions and was conducted between June to September 2021. We found that the psychological and financial impact for citizens and permanent residents stranded abroad due to international travel restrictions was great, with participants reporting moderate to severe levels of anxiety and stress [16]. However, voluntary feedback received from respondents, coupled with

commentary on social media, highlighted that our previous study failed to fully capture the impact and issues relevant to travellers separated from their partners or immediate families, and those unable to enter or return to the country where they hold a temporary visa during the COVID-19 pandemic (henceforth "temporary visa holders". Therefore, building on from that initial study, this subsequent work aimed to evaluate the psychological and financial impacts of those traveller groups by analysing the prevalence of psychological and financial distress, as well as identifying the protective and risk factors associated with specific mental health outcomes of depression, anxiety, and stress.

## Methodology

### Study design and study population

An international cross-sectional survey was designed to evaluate the psychological and financial impact of COVID-19-related travel restrictions. The online open survey was developed using survey tool Qualtrics [17] and respondents were invited to complete the survey voluntarily using an opportunity sampling design, through a variety of Twitter and Facebook group posts which included a call for participants who were eligible based on the clear set of inclusion criteria (see below). These Facebook groups were chosen due to relevant names related to travel/border restrictions, being stranded/stuck abroad, or stranded/stuck overseas. In addition to this, we emailed information about the study and a link to the survey to a previously collected list of impacted individuals who, as part of an effort to count the number of people affected by Australian border closures run by a grassroots advocacy group, opted in to be contacted about future research opportunities regarding separated families and partners and stranded travellers. The online Qualtrics survey took on average 12.25 minutes to complete, gave participants the option to review questions through a back button, and participants were provided with a "not applicable", "none of the above" or "do not recall" option in all questions. Data was collected from 4 November 2021 to 1 December 2021.

Participants were eligible if, due to COVID-19-related travel restrictions, they had experienced separation from an immediate family member or partner OR were a temporary resident unable to enter or return to their country of temporary residence country. A unique IP address was applied to each survey response to prevent duplicate entries. Respondents' anonymity was maintained.

### Ethics approval

Ethical approval for this study was granted by the UNSW Human Research Ethics Committee (#210418). All respondents were provided with a link to the participant information sheet informing participants of the purpose of the study, how the data will be used, the right to withdraw at any time, and information confirming that their data will remain anonymous, participants then indicated their consent to participate by clicking the button to begin the survey.

### Survey instruments and measures

The survey collected data about: sociodemographic characteristics; types and perceived degree of impact of the COVID-19-related travel restrictions; the impact the experience had on their mental wellbeing and their access to psychological support; and the financial impacts of their experience. The self-reporting, 21-item Depression, Anxiety, and Stress Scale (DASS-21) [18] was used to measure three aspects of psychological distress and was selected due to its frequent use in studies of mental health during COVID-19 (19–21). It

**Table 1. Depression anxiety stress scale (DASS 21) score.**

| Symptom severity | Depression | Anxiety | Stress |
|---|---|---|---|
| Normal | 0–9 | 0–7 | 0–14 |
| Mild | 10–13 | 8–9 | 15–18 |
| Moderate | 14–20 | 10–14 | 19–25 |
| Severe | 21–27 | 15–19 | 26–33 |
| Extremely severe | 28+ | 20+ | 34+ |

contains 21 questions assessing the level of symptom severity using a 4-point Likert scale from 0 to 3 (0: 'did not apply to me at all', 1: 'applied to me some of the time', 2: 'applied to me a good part of the time', and 3: 'applied to me most of the time'. Scores in each section are multiplied by 2 to calculate the final score. Using the severity rating key below (Table 1), the resulting scores are categorised as normal, mild, moderate, severe, or extremely severe depression, anxiety, or stress (DAS) [18].

Due to the timeframe of recall while completing the DASS-21 being longer than 2-weeks for some participants, an additional 'do not recall' option was added. Any participants scores who selected 'do not recall' was to be omitted from analysis to maintain validity. Questions within the survey were open and closed ended, with a range of binary questions ('yes/no'), ordinal and nominal scale and Likert scale questions.

Prior to survey rollout, the questionnaire was initially sent for expert reviews by co-authors who have conducted survey studies previously and are experts in the field of public health. Once their feedback was received, questions were either changed for clarity or omitted. A pilot survey was tested among five individuals including co-authors who confirmed the that the target audience would understand the questions being asked. Any confusing or challenging questions were simplified or omitted prior to the roll out, and all results collected were excluded from analysis.

The survey had 36 questions in total, spread out over 13 pages and organised by question subject, with an average of 3.6 questions per page. The completion rate for this questionnaire was 55.7%, and a breakdown of participants who visited the website, attempted the survey, and completed the survey is shown in Fig 1.

## Statistical analysis

Only completed surveys were analysed, incomplete surveys were treated as missing completely at random. Descriptive analysis involved the calculation of means, standard deviations, confidence intervals and standard errors for continuous variable data, and the calculation of counts and proportions for categorical variables data. To test significance, we used one-way ANOVA analyses (for continuous variables) and Chi-square test for associations or a Fishers exact test for smaller cell sizes (for categorical variables). Any variables found to show a significant association with DASS severity ($p < .2$) was chosen to be included in the model as predictor variables. We dichotomised DASS scores to indicate either no-mild symptom severity or moderate-to-extremely severe symptom severity. Finally, we conducted multivariable binary logistic regression to determine demographic, financial, and occupational predictors of moderate to extremely severe depression, anxiety, and stress. Statistical significance was $p < 0.05$. All analyses were conducted with SPSS [19]. No errors, influential outliers or multicollinearity amongst variables was identified. Diagnostics confirmed model assumptions were met.

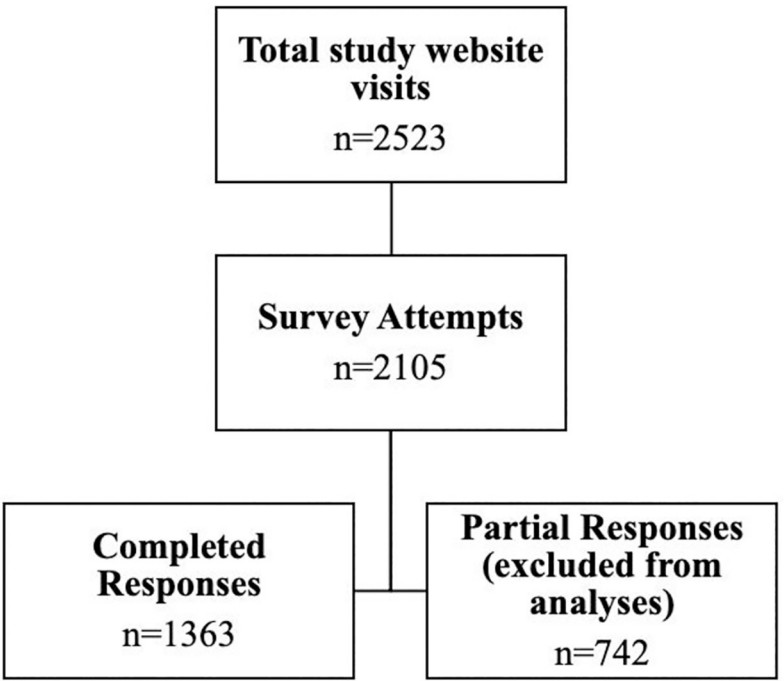

**Fig 1. Flowchart of the participant inclusion process.**

## Results

### Participant characteristics

A total of 1363 respondents completed the survey. Table 2 summarises their sociodemographic characteristics. The mean ± SD age of respondents was 36.7 ± 11.21 years, ranging from 18 to 85 years, with most being female (73.5%, 1002/1363), having a tertiary education (91.9%, 1253/1363) and separated from their immediate family (50.6%, 690/1363).

### Travel experiences

The mean ± SD length of the time stranded, ranged from 1 to 23 months. For participants separated from partners it was 17.3±5.3 months (n = 427), 20.7±3.6 months (n = 690) for those separated from immediate family and 17.1±5.2 months (n = 246) for temporary visa holders. Table 3 summarises the main travel experiences of respondents separated by situation, with those respondents separated from their partners mostly reporting difficulty obtaining entry exemptions (56%, 239/427), entry restrictions without applicable exemptions (51.6%, 220/427) and expensive or unaffordable flights (42.9%, 183/427) as the main concerns. For participants separated from their immediate family, most frequently reported experiences included expensive or unaffordable costs of quarantine (66.4%, 458/690), expensive or unaffordable flights (64.5%, 445/690) and a lack of flight availability (63.8%, 440/690). Finally, for temporary visa holders the most frequently reported experience was entry restrictions with applicable exemptions but difficulty obtaining exemptions (64.2%, 158/246), entry restrictions without applicable exemptions (58.9%, 145/246) and a lack of flight availability (32.1%, 78/246).

**Table 2. Sociodemographic characteristics of temporary visa holders unable to immigrate and those separated from their partner or immediate family during COVID-19.**

| Variables | | *n* | % |
|---|---|---|---|
| **Eligibility** | | **(1363)** | |
| | Separated from immediate family | 690 | 50.6 |
| | Separated from partner/spouse | 427 | 31.3 |
| | Temporary visa holders unable to emigrate | 246 | 18 |
| **Gender** | | **(1363)** | |
| | Woman/Female | 1002 | 73.5 |
| | Man/Male | 348 | 25.5 |
| | Prefer not to say | 10 | .7 |
| | Another term | 3 | .2 |
| **Main language spoken at home** | | **(1363)** | |
| | English | 959 | 70.4 |
| | Other | 339 | 24.9 |
| | Spanish | 36 | 2.6 |
| | French | 18 | 1.3 |
| | Chinese | 9 | .7 |
| | Russian | 2 | .1 |
| **Highest level of education** | | **(1363)** | |
| | Tertiary education | 1253 | 91.9 |
| | Secondary education | 101 | 7.4 |
| | Primary education | 6 | .4 |
| | No formal education | 3 | .2 |
| **Employment during COVID-19** | | **(1363)** | |
| | Other | 626 | 45.9 |
| | Not in paid work | 193 | 45.9 |
| | Essential services worker | 177 | 13 |
| | Health worker | 135 | 9.9 |
| | Educator | 134 | 9.8 |
| | Government worker | 67 | 4.9 |
| | Retired | 31 | 2.3 |
| **Children** | | **(1363)** | |
| | Yes | 823 | 60.4 |
| | No | 540 | 39.6 |
| **Children under 18** | | **(468)** | |
| | 0 | 30 | 6.4 |
| | 1 | 214 | 51.5 |
| | 2 | 145 | 31 |
| | 3 | 40 | 8.5 |
| | 4 or more | 12 | 2.5 |
| **Chronic illness** | | **(1363)** | |
| | Yes | 147 | 10.8 |
| | No | 1174 | 86.1 |
| | Unsure | 42 | 3.1 |

COVID-19, Coronavirus disease of 2019.

**Table 3. Travel experiences of temporary visa holders unable to immigrate and those separated from their partner or immediate family during COVID-19.**

| Variables | | Separated from partner/spouse or immediate family member/s | | |
|---|---|---|---|---|
| | | *n* | % | |
| **Country where participants are/were separated from their partner/spouse or immediate family** | | **(1117)** | | |
| | WPR | 569 | 50.9 | |
| | EUR | 322 | 28.8 | |
| | AMR | 123 | 11 | |
| | EMR | 45 | 4 | |
| | SEAR | 42 | 3.8 | |
| | AFR | 16 | 1.4 | |
| **Country where participants partner or family were/are** | | **(1117)** | | |
| | WPR | 424 | 38.0 | |
| | EUR | 330 | 29.5 | |
| | AMR | 129 | 11.5 | |
| | SEAR | 104 | 9.3 | |
| | EMR | 95 | 8.5 | |
| | AFR | 35 | 3.1 | |
| **Relationship with family members that the participant was/is separated from** | | **n/2014 responses** | **% of participants (690)** | |
| | Parents | 604 | 87.5 | |
| | Aunts/Uncles | 448 | 64.9 | |
| | Siblings | 438 | 63.5 | |
| | Grandparents | 197 | 28.6 | |
| | Children over 18 | 4.6 | 13.9 | |
| | Other | 90 | 13 | |
| | Grandchildren | 41 | 5.9 | |
| | Stepparents | 40 | 5.8 | |
| | Children under 18 | 32 | 4.6 | |
| | Spouse | 21 | 3 | |
| | Stepchildren | 7 | 1 | |
| | | **Temporary visa holders** | | |
| | | n | % | |
| **Country where participants are/were awaiting emigration** | | **(246)** | | |
| | SEAR | 104 | 42.3 | |
| | EMR | 46 | 18.7 | |
| | WPR | 35 | 14.2 | |
| | EUR | 29 | 11.8 | |
| | AMR | 21 | 8.5 | |
| | AFR | 11 | 4.5 | |
| **Country where participants hold a temporary visa and are/were trying to enter or return** | | **(246)** | | |
| | WPR | 216 | 87.8 | |
| | SEAR | 13 | 5.3 | |
| | EUR | 7 | 2.8 | |
| | EMR | 6 | 2.4 | |

(*Continued*)

**Table 3.** (*Continued*)

| | | Separated from partner/spouse or immediate family member/s | | |
|---|---|---|---|---|
| | AMR | 3 | 1.2 | |
| | AFR | 3 | 1.2 | |
| **Initial reason for leaving the country where participants hold a temporary visa** | | **(107)** | | |
| | Visit family and/or friends overseas | 62 | 57.9 | |
| | Other | 33 | 30.8 | |
| | Travel for business reasons | 4 | 3.7 | |
| | Holiday travel | 4 | 3.7 | |
| | Study | 3 | 2.8 | |
| | Travel to study overseas | 1 | .9 | |
| | | **All participants** | | |
| | | **n(%) of Separated partners (427)** | **n(%) of Separated families (690)** | **n(%) of Temporary visa holders (246)** |
| **Travel experiences** | | | | |
| | Lack of flight availability | 178 (41.7) | 440 (63.8) | 78 (32.1) |
| | Expensive or unaffordable flights | 183 (42.9) | 445 (64.5) | 65 (26.4) |
| | Expensive or unaffordable cost of quarantine | 168 (39.3) | 458 (66.4) | 59 (24.0) |
| | Entry restrictions with applicable exemptions at the destination country, but difficulty obtaining the exemption due to processing delays or rejections | 239 (56.0) | 186 (27.0) | 158 (64.2) |
| | Fear of or unwillingness to travel due to risk of border closures, flight cancellations or other external factors | 128 (30.0) | 353 (51.2) | 25 (10.2) |
| | Entry restrictions without applicable exemptions in the destination country | 220 (51.6) | 179 (25.8) | 145 (58.9) |
| | Exit restrictions in the departure country | 147 (34.4) | 305 (44.2) | 16 (6.5) |
| | Lack of time available for travel due to the added time needed for quarantine | 112 (26.2) | 287 (41.6) | 11 (4.5) |
| | Lack of visa due to closed embassy or other processing delays | 137 (32.1) | 64 (9.3) | 26 (10.6) |
| | Lack of knowledge of how to travel in the current global environment | 58 (13.6) | 120 (17.4) | 16 (6.5) |
| | Fear of, or unwillingness to travel due to risk of contracting COVID-19 | 39 (9.1) | 106 (15.4) | 13 (5.3) |
| | Possible to enter the destination country via a third country but cannot because of financial cost, time cots, added risk or ethical reasons | 52 (12.2) | 65 (9.4) | 6 (2.4) |

AFR, African Region; AMR, Region of the Americas; SEAR, South-East Asian Region; EUR, European Region; EMR, Eastern Mediterranean Region; WPR, Western Pacific Region; COVID-19, Coronavirus disease of 2019.

## Mental wellbeing

Mean values and standard deviations for each of the DASS-21 subscales were 11.6±5.9 for depression, 6.8±5.5 for anxiety and 11.5±5.6 for stress. Converted, this translates to 76.8% (1046/1363) meeting the DASS-21 criteria for moderate-to-extremely severe depression, 51.6% (703/1363) between moderate-to-extremely severe anxiety, and 62.6% (853/1363) between moderate-to-extremely severe stress. Fig 2 shows the distribution of DASS-21 mean scores by respondents' situation. The overwhelming majority of respondents reported a lack of

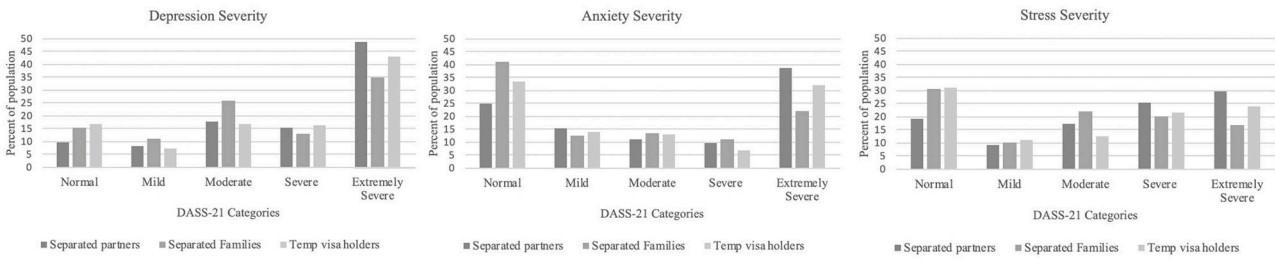

**Fig 2. Distribution of DASS21 severity for participants impacted by COVID-19 related travel restrictions.**

government services offered to them, including mental health services, medical advice, emergency housing, financial assistance and traveller registration being offered to them (see Table 4). Furthermore, 9.4% of respondents reported experiencing homelessness (a point where respondents reported having no shelter in place) while separated or awaiting migration.

## Financial wellbeing

Mean expenditure ranged from $0 to $136,779USD for respondents. For those separated from a partner their average was USD$6078 (SD:11627), an average of USD$5915 (SD:13923) was found for those separated from immediate family, and an average of USD$12,651 (SD: 16036) for temporary visa holders.

Financial stress was reported by 71.2% (971/1363) respondents. Of those separated from a partner/spouse or immediate family member, 64.9% (725/1117) of the respondents' partners' or family members' experienced financial stress. A change in employment status was experienced by 51.4% (700/1363), whereby most experienced having to work remotely (33.3%, 257/1363), followed by loss of employment (25.5%, 197/1363). Table 5 has a full breakdown of the main financial and employment characteristics of respondents.

**Table 4. Government services available to participants during COVID-19.**

| Variables | | *n* | % |
|---|---|---|---|
| **Mental health services available** | | | |
| | Yes | 266 | 19.5 |
| | No | 1097 | 80.5 |
| **Medical advice available** | | | |
| | Yes | 195 | 14.3 |
| | No | 1168 | 85.7 |
| **Emergency housing available** | | | |
| | Yes | 19 | 1.4 |
| | No | 1344 | 98.6 |
| **Government financial assistance available** | | | |
| | Yes | 124 | 9.1 |
| | No | 1239 | 90.9 |
| **Traveler registration available** | | | |
| | Yes | 231 | 16.9 |
| | No | 1132 | 83.1 |

COVID-19, Coronavirus disease of 2019

**Table 5. Financial and employment characteristics of temporary visa holders unable to immigrate and those separated from their partner or immediate family during COVID-19.**

| Variables | | *n* | % |
|---|---|---|---|
| **Additional costs incurred due to delays in emigration or being separated from partner/immediate family** | | n/5138 total responses | % of participants (1363) |
| | Deposits or payments for future travel planned | 638 | 55.3 |
| | COVID-19 tests required for travel | 637 | 55.2 |
| | Application for travel exemption certificates or other COVID related travel procedures | 631 | 54.7 |
| | Cancelled, delayed, changed, or missed flights | 606 | 52.5 |
| | Visa applications | 555 | 48.1 |
| | Immigration lawyers or advisors | 425 | 36.8 |
| | Migration agents | 410 | 35.5 |
| | Notaries, apostille or other legalisation processes | 364 | 31.5 |
| | Mandatory quarantine | 291 | 25.2 |
| | Additional accommodation | 276 | 23.9 |
| | Translators for travel-related procedures | 165 | 14.3 |
| | Other | 140 | 12.1 |
| **Methods of addressing additional financial costs** | | n/1634 total responses | % of participants (1363) |
| | Accessed savings | 777 | 74.7 |
| | Other | 149 | 14.3 |
| | Taken on additional paid employment | 172 | 16.5 |
| | Sold assets (House, investments etc) | 170 | 16.3 |
| | Accessed 'superannuation' or early access to pension funds | 115 | 11.1 |
| | Borrowed from a bank | 108 | 10.4 |
| | Accessed financial support from the social security services | 44 | 4.2 |
| | Applied for a government grant (that you don't have to pay back) for return flights | 44 | 4.2 |
| | Received financial support from your employer | 38 | 3.7 |
| | Applied for a government loan (that you must pay back) for living costs | 21 | 2 |
| | Applied for a government grant (that you don't have to pay back) for living costs | 18 | 1.7 |
| | Accessed financial support from charity services | 6 | 0.6 |
| | Received financial support from an insurance company | 6 | 0.6 |
| | Received financial support from a travel agent | 4 | 0.4 |
| | Applied for a government loan (that you must pay back) for return flights | 3 | .3 |
| **Employment change** | | n/1170 total responses | % of participants (1363) |
| | Work remotely | 257 | 33.3 |
| | Lost job | 197 | 25.5 |
| | Other | 121 | 15.7 |
| | Reduction of hours | 121 | 15.7 |
| | Resigned | 113 | 14.6 |
| | Pay cut | 84 | 10.9 |
| | Stood down, not working for pay, but not fired | 76 | 9.8 |
| | Back in paid work | 59 | 7.6 |
| | Increase in hours | 58 | 7.5 |

*(Continued)*

**Table 5.** (Continued)

| Variables | | *n* | % |
|---|---|---|---|
| | Contract not renewed | 46 | 6 |
| | Not working but still being paid (jobkeeper, furlough) | 38 | 4.9 |

COVID-19, Coronavirus disease of 2019.

## Comparative analysis of depression, anxiety, and stress

There were significant associations between respondents DASS categories and their current situation (separated or temporary visa holder), gender, mental health services offered, medical advice offered, government financial assistance offered, traveller registration offered (significant for depression only), financial stress, homelessness, employment change, having children, chronic illness, expenditure (significant for anxiety and stress only), and respondents' employment during COVID-19. No significant associations were found between DASS severity categories and having emergency accommodation offered, time separated/awaiting immigration (see S1 Table).

For depression, logistic regression identified being female, having mental health services offered, financial stress, and chronic illness as predictors of moderate-to-extremely severe depression. Overall, the model showed goodness of fit to the data ($\chi 2$ (20) = 82.751, p < .001), and correctly discriminated 76.4% of cases.

For anxiety, logistic regression identified being female, having medical advice offered, government financial assistance offered, financial stress, homelessness, chronic illness, and employment during COVID-19, specifically being an essential services worker, as predictors of moderate-to-extremely severe anxiety. Overall, the model showed goodness of fit to the data ($\chi 2$ (20) = 182.342, p < .001), and correctly discriminated 65.9% of cases. Finally for stress, logistic regression identified being separated from a partner, increased expenditure, being female, financial stress, chronic illness, and employment during COVID-19, specifically being an essential services worker or not in paid work, as predictors of moderate-to-extremely severe stress. Overall, the model showed goodness of fit to the data ($\chi 2$ (20) = 166.186, p < .001), and correctly discriminated 67.2% of cases. Being older and having children was associated with decreased odds of moderate-to-extremely severe DAS. Table 6 presents the results of the multivariable logistic regression.

## Discussion

In this study, we examined the prevalence and correlates of psychological and financial distress on those affected by COVID-19-related international travel restrictions and who, as a result, were unable to either, reunite with partners or immediate families, or enter/return to a country where they hold a temporary visa. In comparison to our initial study aimed at citizens and permanent residents stranded abroad in general [16], some differences have emerged between the groups of stranded travellers. The respondents in the current study reported higher values in depression and stress when compared to stranded citizens/permanent residents, who scored 64.2%, 64.4% and 41.7% respectively for depression, anxiety, and stress. Furthermore, when compared to DASS in the published COVID-19 studies and especially in studies focused on migrant workers [20–22], healthcare workers [23] or the general population [24–26], the scores of the present study are dramatically higher, over 50% more severe in some cases. Along with research indicating that those negatively impacted by Australian international border

**Table 6. Predictors of moderate to extremely severe depression, anxiety, and stress in temporary visa holders unable to immigrate and those separated from their partners or immediate family members during COVID-19.**

| Variables | Depression | | | | Anxiety | | | | Stress | | | |
|---|---|---|---|---|---|---|---|---|---|---|---|---|
| | OR | 95% CI | | p | OR | 95% CI | | p | OR | 95% CI | | p |
| **Eligibility** | | | | .273 | | | | .172 | | | | **.002** |
| Separated from partner/spouse* | | | | | | | | | | | | |
| Separated from immediate family | .774 | .556 | 1.077 | | .784 | .595 | 1.033 | .084 | .772 | .578 | 1.031 | |
| Temporary visa holders | .785 | .515 | 1.195 | | .782 | .544 | 1.122 | .182 | .515 | .355 | .748 | |
| **Age** | .983 | .969 | .998 | **.023** | .969 | .957 | .983 | **< .001** | .968 | .955 | .981 | **< .001** |
| **Expenditure** (actual amount/1000) | | | | | 1.037 | .974 | 1.103 | .255 | 1.104 | 1.028 | 1.186 | **.007** |
| **Gender** | | | | | | | | | | | | |
| Male* | | | | | | | | | | | | |
| Female | 1.469 | 1.065 | 2.026 | **.019** | 1.691 | 1.265 | 2.260 | **< .001** | 1.575 | 1.176 | 2.110 | **.002** |
| **Mental health services offered (ref. no)** | 1.592 | 1.026 | 2.471 | **.038** | .934 | .656 | 1.330 | .706 | 1.357 | .936 | 1.968 | .108 |
| **Medical advice offered (ref. no)** | 1.214 | .731 | 2.017 | .454 | 2.159 | 1.424 | 3.274 | **< .001** | 1.444 | .935 | 2.232 | .098 |
| **Government financial assistance offered (ref. no)** | 1.134 | .688 | 1.868 | .621 | 1.568 | 1.025 | 2.399 | **.038** | 1.199 | .769 | 1.870 | .422 |
| **Traveler registration offered (ref. no)** | .800 | .565 | 1.132 | .208 | | | | | | | | |
| **Financial Stress (ref. no)** | 1.867 | 1.393 | 2.502 | **< .001** | 2.246 | 1.713 | 2.946 | **< .001** | 2.115 | 1.615 | 2.772 | **< .001** |
| **Homelessness (ref. no)** | 1.158 | .712 | 1.882 | .555 | 1.837 | 1.209 | 2.791 | **.004** | 1.136 | .739 | 1.747 | .562 |
| **Employment change (ref. no)** | .900 | .682 | 1.186 | .453 | .939 | .737 | 1.197 | .612 | 1.151 | .898 | 1.475 | .267 |
| **Children (ref. no)** | .694 | .514 | .939 | **.018** | .735 | .561 | .963 | **.025** | .792 | .602 | 1.041 | .094 |
| **Chronic Illness (ref. no)** | 1.707 | 1.054 | 2.765 | **.030** | 1.662 | 1.133 | 2.437 | **.009** | 2.111 | 1.384 | 3.220 | **< .001** |
| **Employment during COVID-19** | | | | .188 | | | | **.006** | | | | **.042** |
| Healthcare worker | .816 | .527 | 1.263 | | .743 | .496 | 1.113 | | .809 | .539 | 1.213 | |
| Government worker | 1.939 | .965 | 3.896 | | 1.497 | .862 | 2.600 | | 1.222 | .695 | 2.146 | |
| Essential services | 1.363 | .884 | 2.103 | | 1.755 | 1.755 | 1.215 | | 1.522 | 1.035 | 2.239 | |
| Educator | .973 | .615 | 1.538 | | 1.023 | 1.023 | .681 | | .970 | .640 | 1.471 | |
| Not in paid work | 1.219 | .804 | 1.849 | | 1.296 | 1.296 | .906 | | 1.533 | 1.054 | 2.229 | |
| Retired | 1.802 | .717 | 4.529 | | 2.188 | 2.188 | .896 | | 1.893 | .792 | 4.525 | |
| Other * | | | | | | | | | | | | |

*Reference variable

OR, log odds ratio controlling for other variables in the model; COVID-19, Coronavirus disease of 2019; EXP (B), adjusted odds ratio; C.I, confidence interval; p, probability value (statistically significant < .05).

*Notes*. Only variables found to have been statistically significant (< .2) from the chi-square analyses were included in each model

closures were at an increased risk of elevated psychological distress [27], these scores combined provide additional evidence that travel restrictions have negative psychological consequences towards those directly impacted by them.

As direct social contact has been shown to be a protective factor for psychological distress [25], it may explain why our respondents had much higher scores compared to the other populations, as they were unable to be with friends or family, either directly because of travel restrictions or indirectly due to quarantine or lockdown. Research suggests that experiences that are distressing and demanding and which occur every day for longer than 6 months are considered a chronic psychological stressor which has been linked to early disease conditions and mortality [28]. Given the impact of social isolation, loneliness and financial stress on mental health, and projected increase in suicide and suicide ideation reported in research from Canada, USA, Pakistan, India, France and many others, these findings are particularly concerning [29–31]. Results from this study can be used to assist in defining groups who are at an

increased risk of developing moderate-to-extremely severe depression, anxiety, and stress due to international travel restrictions.

Our data suggests a significant financial burden on those impacted by travel restrictions, with respondents' average expenditure incurred $7,285USD and 71.2% reporting financial stress. These results are higher than those of our previous study [16] which reported financial stress in 64.2% and 45% experience an employment change. This difference could be due to the increased financial insecurity and visa issues/costs experienced by migrant workers, already evident in previous research [11, 12]. Additionally, the respondents in this study reported unique additional costs of visa applications, travel certificates, cancelled or delayed flights, visa applications, migration agents and lawyers to name a few. Interestingly, we found that 64.9% of the respondents' partners' or family members' experienced financial stress due to the forced separation. This result warrants further research into the impacts of travel restrictions more broadly.

Of those that experienced a change in employment, 25.5% reported losing their job while separated from loved ones or awaiting immigration. This result is consistent with previous research during COVID-19 [32–34], with a study in South Africa reporting job loss rates of 30% [34] and a study on mothers and children experiencing adversity in Australia reporting 27% job and income loss [32]. Previous literature suggests that experiences of employment changes and high additional expenditure can increase feelings of financial distress which in turn is likely to exacerbate psychological distress [34–36]. Our findings reflect this with results showing financial distress caused by travel restrictions to be a statistically significant predictor of moderate-to-extremely severe DAS.

Our findings regarding respondents' experiences and perceptions towards the degree of government support reflects a very bleak picture, with almost all reporting that there was no support available, including no available mental health service (80.5%), no medical advice available (85.7%), no emergency housing available (98.6%), no government financial assistance available (90.9%) and finally no traveller registration available (83.1%). These results, along with the varied support and services available to citizens stranded abroad evident in our previous study, especially that of mental health and emergency housing [37], reflect the ongoing inequity in access and availability of healthcare, resources, and services [38]. Due to a perceived lack of government support along with high reported incidence of DASS severity and financial stress, we strongly recommend policymakers consider and mitigate the negative impacts of restricting temporary visa holders or immediate family members and partners from entering countries for reunification or migration purposes. Furthermore, we recommend policymakers introduce additional proactive support to those impacted by travel restrictions, specifically increasing the access and availability of community services, social support through mental health interventions and government financial aid, especially for immigration and visa support.

In line with results from our previous study and that of other research during COVID-19, women and those with chronic illness were found to be significantly associated with higher severity of DAS [25, 39, 40]. These results are consistent with evidence showing women and those with chronic illness to be more vulnerable to psychological problems than males and those without chronic illness [25, 41]. Current evidence indicates those with chronic illnesses had increased fatality rates and disproportional effects during COVID-19, which could further exacerbate psychological distress linked to separation from loved ones or awaiting migration or travel [42, 43].

Finally, results indicate that a respondent's situation (i.e. whether they were separated from family, separated from partner/spouse or delayed migration or travel on a temporary visa), was not the significant predictor of moderate to extremely severe depression or anxiety. This

result suggests that for the most part, the negative psychological impacts of international travel restrictions did not vary depending on participant situation, and therefore evidence from this study provides a holistic picture of the psychological impacts felt by the many individuals impacted by travel restriction. However, it is also important to note, that we did not set the sample size up to compare between groups and so we may not have been sufficiently powered.

## Limitations

This study has several limitations. Firstly, the results are based on a cross-sectional survey design, and lacks longitudinal follow up. Most respondents were female (73.5%), educated to a tertiary level (91.9%), with an ability to read English and exhibit a competent level of computer literacy due to the online survey format, which could introduce sampling bias as it may not be fully representative of those impacted by international travel restrictions globally who may not speak English or who are lack computer literacy. Secondly, while we used a validated tool, our psychometric results were based on self-reporting, and as such could raise some subjectivity and bias issues as the scores reflect psychological distress not diagnosed by mental health professionals. Further, a baseline measure of mental health was not available or collected, and hence measurement in change due to COVID-19-related travel restrictions was not possible. Despite these limitations, the study provides novel insights into the wider impacts of travel restrictions beyond infection control.

## Conclusion

Our findings suggest that international travel restrictions have significantly contributed to severe symptoms of depression, anxiety, and stress, as well as financial distress. Results indicate that for those impacted by COVID-19 related travel restrictions, being female, chronic illness and financial stress were predictors of moderate to extremely severe DAS. Work status during COVID-19, specifically being an essential services worker or unemployed were both predictors of higher anxiety, with only essential workers being a predictor of higher stress severity. Being older and having children was associated with decreased odds of moderate to extremely severe DAS. Most respondents reported no government support being offered. The results of this study show both the psychological and financial impacts of international travel restrictions, as well as a severe gap in government services available that are vital to assisting vulnerable groups.

## Supporting information

**S1 Table. Associations and variance between participant factors and levels of depression, anxiety, and stress (n = 1363).**
(DOCX)

## Acknowledgments

Thank you to the participants for completing the survey.

## Author Contributions

**Conceptualization:** Pippa McDermid, Soumya Sooppiyaragath, Adam Craig, Meru Sheel, Holly Seale.

**Data curation:** Pippa McDermid, Meru Sheel, Holly Seale.

**Formal analysis:** Pippa McDermid, Katrina Blazek, Holly Seale.

**Investigation:** Pippa McDermid, Siobhan Talty.

**Methodology:** Pippa McDermid, Soumya Sooppiyaragath, Adam Craig, Meru Sheel, Siobhan Talty, Holly Seale.

**Project administration:** Holly Seale.

**Supervision:** Holly Seale.

**Writing – original draft:** Pippa McDermid, Holly Seale.

**Writing – review & editing:** Pippa McDermid, Soumya Sooppiyaragath, Adam Craig, Meru Sheel, Katrina Blazek, Siobhan Talty, Holly Seale.

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
