## [Decision Letter · Decision Letter 0]

14 Mar 2022

PONE-D-22-05305Psychological and financial impacts of COVID-19-related travel measures: An international cross-sectional studyPLOS ONE

Dear Dr. Seale,

Thank you for submitting your manuscript to PLOS ONE. After careful consideration, we feel that it has merit but does not fully meet PLOS ONE’s publication criteria as it currently stands. Therefore, we invite you to submit a revised version of the manuscript that addresses the points raised during the review process.

We look forward to receiving your revised manuscript.

Kind regards,

Latika Gupta

Academic Editor

PLOS ONE

Journal Requirements:

Additional Editor Comments:

Please detail methods using the cHERRIES checklist.

Please describe survey validation and pilot testing in detail.

An infographic to summarise design ad results and further directions would be welcome.

Please discuss sampling bias as a major limitation.

Reviewers' comments:

Reviewer's Responses to Questions

**Comments to the Author**

1. Is the manuscript technically sound, and do the data support the conclusions?

Reviewer #1: Yes

2. Has the statistical analysis been performed appropriately and rigorously? 

Reviewer #1: Yes

3. Have the authors made all data underlying the findings in their manuscript fully available?

Reviewer #1: Yes

4. Is the manuscript presented in an intelligible fashion and written in standard English?

Reviewer #1: Yes

5. Review Comments to the Author

Reviewer #1: The authors examined the prevalence and correlation of psychological and financial distress on those affected by COVID-19-related international travel restrictions and who, as a result, were unable to either, reunite with partners or immediate families, or enter/return to a country where they hold a temporary visa.

The major limit of the study as assed by the authors was the absence of a baseline measure of mental health.

Minors comments:

Objectives of the study should be added in the end of the introduction section

How did you construct and validate the survey? Did you do a pilot test before launching it?

Footnotes in tables with abbreviations are lacking

Figure 1 need to be improved

Table 5: you could remove the p column to be more efficient.

6. PLOS authors have the option to publish the peer review history of their article (what does this mean?). If published, this will include your full peer review and any attached files.

Reviewer #1: No

---

## [Author Response · Author response to Decision Letter 0]

4 Apr 2022

Editors Comments

a. Titles and file names have been changed to reflect PLOS ONE’s style requirements

2. Please provide additional details regarding participant consent. In the ethics statement in the Methods and online submission information, please ensure that you have specified what type you obtained (for instance, written or verbal, and if verbal, how it was documented and witnessed). If your study included minors, state whether you obtained consent from parents or guardians. If the need for consent was waived by the ethics committee, please include this information. Once you have amended this/these statement(s) in the Methods section of the manuscript, please add the same text to the “Ethics Statement” field of the submission form (via “Edit Submission”).

a. The ethics statement within the methods section has been updated to include the following:

“All respondents were provided with a link to the participant information sheet informing participants of the purpose of the study, how the data will be used, the right to withdraw at any time, and information confirming that their data will remain anonymous, participants then indicated their consent to participate by clicking the button to begin the survey.”

a. We have amended the data availability statement 

4. Please review your reference list to ensure that it is complete and correct. If you have cited papers that have been retracted, please include the rationale for doing so in the manuscript text, or remove these references and replace them with relevant current references. Any changes to the reference list should be mentioned in the rebuttal letter that accompanies your revised manuscript. If you need to cite a retracted article, indicate the article’s retracted status in the References list and include a citation and full reference for the retraction notice.

a. The reference list has been reviewed and it complete and correct. 

5. Please detail methods using the cHERRIES checklist.

a. The methods have been updated to include cHERRIES checklist criteria.

6. Please describe survey validation and pilot testing in detail.

a. The methods have been updated to reflect survey validation. 

“Prior to survey rollout, the questionnaire was initially sent for expert reviews by co-authors who have conducted survey studies previously and are experts in the field of public health. Once their feedback was received, questions were either changed for clarity or omitted. A pilot survey was tested among five individuals including co-authors who confirmed the that the target audience would understand the questions being asked. Any confusing or challenging questions were simplified or omitted prior to the roll out, and all results collected were excluded from analysis.”

7. Please discuss sampling bias as a major limitation.

a. Limitations adjusted to reflect sampling bias as a major limitation:

“Most respondents were female (73.5%), educated to a tertiary level (91.9%), with an ability to read English and exhibit a competent level of computer literacy due to the online survey format, which could introduce sampling bias as it may not be fully representative of those impacted by international travel restrictions globally who may not speak English or who lack the capacity to complete the survey online.”

Reviewer 1

1. Objectives of the study should be added in the end of the introduction section

a. Introduction has been edited to address aims and objectives:

“Therefore, building on from that initial study, this subsequent work aimed to evaluate the psychological and financial impacts of those traveller groups by analysing the prevalence of psychological and financial distress, as well as identifying the protective and risk factors associated with specific mental health outcomes of depression, anxiety, and stress.”

2. How did you construct and validate the survey? Did you do a pilot test before launching it?

a. The methods have been updated to reflect survey validation: 

“Prior to survey rollout, the questionnaire was initially sent for expert reviews by co-authors who have conducted survey studies previously and are experts in the field of public health. Once their feedback was received, questions were either changed for clarity or omitted. A pilot survey was tested among five individuals including co-authors who confirmed the that the target audience would understand the questions being asked. Any confusing or challenging questions were simplified or omitted prior to the roll out, and all results collected were excluded from analysis.”

3. Footnotes in tables with abbreviations are lacking

a. The footnotes in ALL table have been updated.

4. Figure 1 need to be improved

a. Colour and font have been adjusted to comply with PLOS ONE’s figure guide

5. Table 5: you could remove the p column to be more efficient.

a. We find value in including the p column, however sub-variable p values have been removed for efficiency

---

## [Decision Letter · Decision Letter 1]

18 May 2022

PONE-D-22-05305R1Psychological and financial impacts of COVID-19-related travel measures: An international cross-sectional studyPLOS ONE

Dear Dr. Seale,

Thank you for submitting your manuscript to PLOS ONE. After careful consideration, we feel that it has merit but does not fully meet PLOS ONE’s publication criteria as it currently stands. Therefore, we invite you to submit a revised version of the manuscript that addresses the points raised during the review process.

We look forward to receiving your revised manuscript.

Kind regards,

Latika Gupta

Academic Editor

PLOS ONE

Journal Requirements:

Additional Editor Comments (if provided):

Dear authors, please improve methods with reference to standard process of conducting survey based studies e.g. CHERRIES checklist, 10.3346/jkms.2020.35.e398

Reviewers' comments:

Reviewer's Responses to Questions

**Comments to the Author**

1. If the authors have adequately addressed your comments raised in a previous round of review and you feel that this manuscript is now acceptable for publication, you may indicate that here to bypass the “Comments to the Author” section, enter your conflict of interest statement in the “Confidential to Editor” section, and submit your "Accept" recommendation.

Reviewer #2: (No Response)

Reviewer #3: All comments have been addressed

2. Is the manuscript technically sound, and do the data support the conclusions?

Reviewer #2: Partly

Reviewer #3: Yes

3. Has the statistical analysis been performed appropriately and rigorously? 

Reviewer #2: No

Reviewer #3: Yes

4. Have the authors made all data underlying the findings in their manuscript fully available?

Reviewer #2: Yes

Reviewer #3: (No Response)

5. Is the manuscript presented in an intelligible fashion and written in standard English?

Reviewer #2: Yes

Reviewer #3: Yes

6. Review Comments to the Author

Reviewer #2: 1-"Using a key, the resulting scores are categorised as normal,

mild, moderate, severe, or extremely severe depression, anxiety, or stress (DAS)." Relevant reference is required; please also state the cut-offs.

2-"Only completed surveys were analysed." There are ways of dealing with missing data.

3-"Chi-square analyses " which chi-squared test was used?

4-How was the normality of data checked?

5-Manufacturer and country of origin should be given for statistical software.

6-Which program was used for online survey?

7-"Statistical significance was p > 0.05." Please recheck.

8-How was the sample size arrived at?

9-What was the response rate?

10-Any statistical correction for multiple comparisons?

Reviewer #3: The process of selection of the study participants was not very clear. Authors spoke about a previous study but the current method of selection could be explained better.

Page 3 last paragraph- 2nd line - " forced separation" instead of " forced separated".

Could the bias of relatively high number of female respondents be resolved by using weighted values?

7. PLOS authors have the option to publish the peer review history of their article (what does this mean?). If published, this will include your full peer review and any attached files.

Reviewer #2: No

Reviewer #3: **Yes: **Tulika Chatterjee, MD, MPH

---

## [Author Response · Author response to Decision Letter 1]

17 Jun 2022

Journal Requirements

Reference for the initial study has been updated with official publication citation (16).

Editor comments

Please improve methods with reference to standard process of conducting survey based studies e.g. CHERRIES checklist, 10.3346/jkms.2020.35.e398

Added missing ‘Cherries checklist’ details in methods related to – Sampling design, response rate diagram and handling of incomplete questions.

Reviewer comments

Reviewer 2

1. "Using a key, the resulting scores are categorised as normal,

mild, moderate, severe, or extremely severe depression, anxiety, or stress (DAS)." Relevant reference is required; please also state the cut-offs.

a. A table has been created to reflect cut off scores and a reference has been added

2. "Only completed surveys were analysed." There are ways of dealing with missing data.

a. Of the respondents who did complete questions on the outcome (DAS score), most did not complete the last page of the survey which contained all questions related to demographic information. We considered the use of multiple imputation however these methods are best suited when missing information is sparse. In our scenario, participants were missing all demographic information. We have compared the outcomes for those with and without missing demographic information and found the distribution of scores to be comparable. Therefore, we do not have a reason to believe that there are meaningful differences between the those with and without missing demographic information and are treating them as missing completely at random (Methods under “Statistical Analysis” has been updated to reflect this). A third group of respondents began the survey but did not complete questions related to outcome. As imputation methods are best used for predictor variables, we did not include any of these responders in the analysis. A new diagram has been created to reflect the breakdown of completed responses - Figure 1. Flowchart of the participant inclusion process (2).tiff.

3. "Chi-square analyses " which chi-squared test was used?

a. Sentence has been updated to reflect the use of chi square test for associations

4. How was the normality of data checked?

a. Continuous outcomes were visually assessed for normality. Methods has been updated to reflect this “No errors, influential outliers or multicollinearity amongst variables was identified. Diagnostics confirmed model assumptions were met.”

5. Manufacturer and country of origin should be given for statistical software.

a. Citation for SPSS guided by: https://www.ibm.com/support/pages/how-cite-ibm-spss-statistics-or-earlier-versions-spss

6. Which program was used for online survey?

a. Discussed and referenced in ‘Study design and study population’ section - “The online open survey was developed using survey tool Qualtrics”

7. "Statistical significance was p > 0.05." Please recheck.

a. Thank you for picking this up. The sentence has been corrected to p < 0.05

8. How was the sample size arrived at?

a. A minimum sample size of 385 calculated to allow for a proportion of 50% reporting moderate to severe DAS scores with a margin of error of 5%

9. What was the response rate?

a. Due to the study design using opportunistic sampling, the response rate cannot be determined.

10. Any statistical correction for multiple comparisons?

a. The results of the regression analyses are considered to be hypothesis generating, therefore corrections for multiple comparisons have not been applied.

Reviewer 3

The process of selection of the study participants was not very clear. Authors spoke about a previous study but the current method of selection could be explained better.

Page 3 last paragraph- 2nd line - " forced separation" instead of " forced separated".

Could the bias of relatively high number of female respondents be resolved by using weighted values?

The opportunistic sampling strategy precludes the identification of the sampling frame, therefore we are unable to apply weighting methods. We have accounted for potential confounding by the inclusion of sex and other important demographic variables in the regression models.

---

## [Decision Letter · Decision Letter 2]

11 Jul 2022

Psychological and financial impacts of COVID-19-related travel measures: An international cross-sectional study

PONE-D-22-05305R2

Dear Dr. Seale,

We’re pleased to inform you that your manuscript has been judged scientifically suitable for publication and will be formally accepted for publication once it meets all outstanding technical requirements.

Kind regards,

Latika Gupta

Academic Editor

PLOS ONE

Additional Editor Comments (optional):

-

Reviewers' comments:

Reviewer's Responses to Questions

**Comments to the Author**

1. If the authors have adequately addressed your comments raised in a previous round of review and you feel that this manuscript is now acceptable for publication, you may indicate that here to bypass the “Comments to the Author” section, enter your conflict of interest statement in the “Confidential to Editor” section, and submit your "Accept" recommendation.

Reviewer #1: (No Response)

Reviewer #3: All comments have been addressed

2. Is the manuscript technically sound, and do the data support the conclusions?

Reviewer #1: Yes

Reviewer #3: Yes

3. Has the statistical analysis been performed appropriately and rigorously? 

Reviewer #1: Yes

Reviewer #3: Yes

4. Have the authors made all data underlying the findings in their manuscript fully available?

Reviewer #1: Yes

Reviewer #3: Yes

5. Is the manuscript presented in an intelligible fashion and written in standard English?

Reviewer #1: Yes

Reviewer #3: Yes

6. Review Comments to the Author

Reviewer #1: (No Response)

Reviewer #3: This is the second revision of the paper I am reviewing, authors have address all the queries and questions raised.

7. PLOS authors have the option to publish the peer review history of their article (what does this mean?). If published, this will include your full peer review and any attached files.

Reviewer #1: No

Reviewer #3: No

---

## [Editor Report · Acceptance letter]

19 Jul 2022

PONE-D-22-05305R2 

Psychological and financial impacts of COVID-19-related travel measures: An international cross-sectional study 

Dear Dr. Seale:

I'm pleased to inform you that your manuscript has been deemed suitable for publication in PLOS ONE. Congratulations! Your manuscript is now with our production department. 

Kind regards, 

on behalf of

Dr. Latika Gupta 

Academic Editor

PLOS ONE